# AXR1 affects DNA methylation independently of its role in regulating meiotic crossover localization

Nicolas Christophorou[1], Wenjing She[2], Jincheng Long[2], Aurélie Hurel[1], Sébastien Beaubiat[1], Yassir Idir[1], Marina Tagliaro-Jahns[1¤a], Aurélie Chambon[1], Victor Solier[1¤b], Daniel Vezon[1], Mathilde Grelon[1], Xiaoqi Feng[2], Nicolas Bouché[1☯*], Christine Mézard[1☯*]

1 Institut Jean-Pierre Bourgin, UMR1318 INRA-AgroParisTech, Université Paris-Saclay, Versailles, France,
2 Department of Cell and Developmental Biology, John Innes Centre, Norwich, United Kingdom

☯ These authors contributed equally to this work.
¤a Current address: Cytogenetics Department, Sullivan Nicolaides Pathology, Brisbane, Australia
¤b Current address: Department of Chromosome Biology, Max Planck Institute for Plant Breeding Research (MPIPZ), Carl-von-Linne Weg, Cologne, Germany
* nicolas.bouche@inrae.fr (NB); christine.mezard@inrae.fr (CM)

**Data Availability Statement:** Methylome genome wide data have been deposited to the ENA database under the accession number PRJEB28434.

## Abstract

Meiotic crossovers (COs) are important for reshuffling genetic information between homologous chromosomes and they are essential for their correct segregation. COs are unevenly distributed along chromosomes and the underlying mechanisms controlling CO localization are not well understood. We previously showed that meiotic COs are mis-localized in the absence of AXR1, an enzyme involved in the neddylation/rubylation protein modification pathway in *Arabidopsis thaliana*. Here, we report that in *axr1^-/-*, male meiocytes show a strong defect in chromosome pairing whereas the formation of the telomere bouquet is not affected. COs are also redistributed towards subtelomeric chromosomal ends where they frequently form clusters, in contrast to large central regions depleted in recombination. The CO suppressed regions correlate with DNA hypermethylation of transposable elements (TEs) in the CHH context in *axr1^-/-* meiocytes. Through examining somatic methylomes, we found *axr1^-/-* affects DNA methylation in a plant, causing hypermethylation in all sequence contexts (CG, CHG and CHH) in TEs. Impairment of the main pathways involved in DNA methylation is epistatic over *axr1^-/-* for DNA methylation in somatic cells but does not restore regular chromosome segregation during meiosis. Collectively, our findings reveal that the neddylation pathway not only regulates hormonal perception and CO distribution but is also, directly or indirectly, a major limiting pathway of TE DNA methylation in somatic cells.

## Author summary

In sexually reproducing organisms, each parent transmits one and only one copy of each chromosome to their progeny via their packaging in haploid gametes. To ensure the proper transmission of the chromosomes, pairs of homologous chromosomes must

**Funding:** This work was supported by the Department of Biology and Amelioration des Plantes of the Institut National de Recherche Agronomique, Projet incitatif [BAP2013_4_ANACOVA] and LabEx Saclay Plant Sciences, [SPS-Research Open Call 2014_CHROMAREC]. The Institut Jean-Pierre Bourgin benefits from the support of the LabEx Saclay Plant Sciences [Project 10-LABX-0040-SPS]. Wenjing She, Jincheng Long and Xiaoqi Feng are supported by a BBSRC Responsive Mode grant [BB/S009620/1; Wenjing She and Xiaoqi Feng] and a European Research Council Starting grant [SexMeth; Jincheng Long and Xiaoqi Feng] and a BBSRC Institute Strategic Programme [GEN; BB/P013511/1; Xiaoqi Feng]. The funders had no role in study design, data collection and analysis, decision to publish, or preparation of the manuscript.

**Competing interests:** The authors have declared that no competing interests exist.

associate and exchange genetic information (also called reciprocal recombination) during a special division called meiosis that lead to the formation of the gametes. The recombination process is highly controlled in terms of number and localization of the events along the chromosomes. Disruption of this control may cause an inappropriate transmission of the chromosomes in the gametes leading to abnormal chromosome numbers in the offspring which is usually deleterious. In the plant *Arabidopsis thaliana*, we show that when the pathway modifying proteins through ubiquitination/neddylation is impaired, the number of reciprocal recombination events is maintained but they are delocalized toward the ends of the chromosomes and some chromosomes do not exchange material. We also detected changes of patterns for DNA methylation, an epigenetic modification localised on DNA cytosines. Furthermore, we demonstrate that the methylation of cytosines is not causal to the localization change of meiotic recombination events.

## Introduction

Meiosis is a specialized type of cell division that produces haploid spores that eventually develop into gametes. It requires a single round of DNA replication followed by two successive rounds of chromosome segregation. One key feature of the first meiotic division is the reciprocal exchange of genetic material also called a crossover (CO), which is one of the outcomes of meiotic recombination. COs are formed by the repair of programmed meiotic DNA double strand breaks (DSBs). DSBs are produced at the onset of meiosis generally in a large excess compared to the number of COs formed. The proportion of DSB precursors converted into COs varies from around two thirds in *Saccharomyces cerevisiae* to a few percent in mammals and plants [1]. Two different pathways contribute to CO formation: class I COs depend on the ZMM proteins (Zip1, Zip2, Zip3, Zip4, Msh4, Msh5, Spo16 and Mer3), in addition to Mlh1 and Mlh3. Their distribution is affected by interference (adjacent COs are more regularly spaced than expected if they were randomly distributed [2]). Class II COs depend notably on the Mus81/Eme1 protein complex and do not interfere (reviewed in [3]).

In many species, COs are essential for the accurate segregation of homologous chromosomes at the first meiotic division. Each pair of chromosomes receives at least one CO, the so-called obligatory CO [4]. COs, together with sister chromatid cohesion, mediate the physical association of homologous chromosomes into bivalents. In the absence of CO formation, homologous chromosomes segregate randomly leading to aneuploid gametes that are either unviable or affect the viability or development of offspring (reviewed in [3]). Moreover, COs are a driving force in evolution, generating novel combinations of alleles on which selection can act.

In most species, CO distribution is not homogeneous along the genome: domains with higher and lower CO rates than the genome average alternate along the chromosomes. One universal observation is that centromeres and centromere proximal regions (pericentromeres) are suppressed for CO formation [5]. Centromeres are defined as regions where the kinetochores assemble as the centromere specific Cen-H3/CENP-A histone variant is deposited [6]. The kinetochore is the major factor responsible for setting up a repressive environment for CO recombination during meiosis [7]. In most multicellular organisms, pericentromeres are compacted heterochromatic regions dense in transposable elements (TEs) and repetitive sequences, characterized by high levels of methylation on both DNA and lysine 9 of histone H3 (H3K9) [8]. CO suppression in pericentromeric regions is particularly marked in crop species with large genomes where pericentromeres can occupy more than half of the

chromosomes. Contrary to original assumptions, these CO-poor heterochromatic regions are not devoid of genes [9,10]. In barley for example, 48% of the 5.1 Gb genome assigned to centromeric and pericentromeric regions contains as much as 22% of the total gene content but nevertheless exhibits drastically reduced recombination frequency [11]. The origin of such chromosome heterogeneity in recombination is still poorly understood, but several lines of evidence argue that the shape of CO distribution is the result of multilayer controls that could be interconnected and vary from species to species [12]. First, DSB distribution along the chromosomes may influence the CO map. Indeed, the CO landscape appears to mirror significantly, albeit not entirely, the DSB site map in some vertebrates such as human and mice [13,14]. However, in maize the CO distribution does not follow the distribution of DSBs but the subset of genic DSBs correlates more strongly with CO localization [15]. In *Arabidopsis*, a positive relationship between DSBs and CO has been reported genome wide, but at a fine scale DSBs are detected in CO hotspot regions but not at a significantly higher level than in randomly positioned loci [16]. Spatio-temporal regulation of recombination initiation (DSB formation) can also have an impact on CO formation. DSB formation is temporally and spatially coordinated with DNA replication (17). Correspondingly, in *S. cerevisiae* or in barley, DSBs appear gradually along the genome [17,18]. In *S. cerevisiae*, the CO/DSB ratio varies according to chromosomal location [19], time of induction and/or level of DSBs [20]. In both human and barley male meiosis, precocious recombination initiation in subtelomeric chromosomal regions correlates with subtelomeric CO formation in these regions [18,21].

In addition to DSB distribution, several lines of evidence point towards a major contribution of the axial element (AE) of the chromosomes in controlling CO distribution. Chromosomes display a specific structure during meiotic prophase I, with sister chromatid loops anchored on the AE and with variation in chromosome axis length inversely correlating with variations in chromatin loop sizes [22]. In tomato, mouse and *Caenorhabditis elegans*, artificial depletion or defects in the dynamics of components of the AE change the CO distribution [23–25]. An additional level of control is meiotic chromosomal dynamics. Meiotic prophase I is accompanied by a number of spectacular structural changes within the nucleus that include telomere bouquet formation, chromosome movement, chromosome pairing, and synapsis. Chromosome movement during meiotic prophase is a common feature in eukaryotes and appears to be an important regulator of meiotic recombination [26]. In *Arabidopsis*, SUNs proteins [27] and PSS1 [28], a Kinesin1-like protein, that are homologs of proteins known to be part of the chromosome mobility machinery in yeast, worms and mammals, are both required for meiotic CO control.

In *Arabidopsis*, different pathways methylate DNA in the three following sequence contexts: CG, CHG and CHH, where H can be A, T or C [29]. Cytosines can be methylated *de novo*, in all contexts, by an RNA-directed DNA methylation (RdDM) mechanism involving plant specific DNA-dependent RNA polymerases (Pol IV and Pol V) and the production of small interfering RNAs [30]. DNA METHYLTRANSFERASE 1 (MET1) maintains CG methylation genome-wide, within both TEs and gene bodies. TE silencing is maintained by both the RdDM pathway and a positive reinforcing loop between non-CG methylated DNA and histones. Indeed, CHROMOMETHYLASES (CMTs), like CMT3, are DNA methyltransferases maintaining non-CG methylation and recruited to regions enriched in methylation on lysine 9 of histone H3 (H3K9). Reciprocally, H3K9 histone methyltransferases bind cytosines methylated at non-CG sites to methylate the associated histones [29]. In many species, reprogrammation of DNA methylation occurs in various tissues and cellular lineages and appears to be a common regulatory mechanism in various tissues. In *Arabidopsis* male meiocytes, transposons have high levels of CG and CHG methylation but a lower CHH methylation level compared to somatic tissues [31]. Several studies have reported that disrupting DNA methylation pathway

is associated with some changes in the distribution of CO along the chromosomes. For example, inhibiting either the CHG maintaining pathway, mediated by CMT3, or H3K9 methylation, results in a slight but significant increase in *Arabidopsis* CO formation in pericentromeres with simultaneous moderate reduction of CO formation in chromosome arms [32]. Moreover, in *Arabidopsis met1* mutants, where CG methylation is reduced within centromeres and surrounding regions, the number of COs increases in chromatin arms and decreases in pericentromeric regions [33–36]. However, in these studies, methylome sequencings were performed in somatic tissues and the patterns of methylation in meiocytes remains unknown, highlighting that the interplay between DNA methylation and CO localization is not yet fully understood.

We previously reported that a mutation in *Arabidopsis AXR1*, involved in the neddylation pathway, resulted in mislocalization of class I COs [37], without reducing their overall number. Here, we show that COs cluster in distal regions of the chromosomes in $axr1^{-/-}$, while large central regions of the chromosomes are cold for CO formation. The CO depleted regions correlate with DNA hypermethylation of TEs in the CHH context in $axr1^{-/-}$ meiocytes. In somatic cells, we also observed DNA hypermethylation for $axr1^{-/-}$ TEs in all cytosine contexts, highlighting the strong impact of $axr1^{-/-}$ on the DNA methylome in general. Inactivating of the DNA methylation pathways in $axr1^{-/-}$ reverts the DNA hypermethylation of somatic cells but not the $axr1^{-/-}$ meiotic defects.

## Results

### Genome-wide analysis of male meiotic recombination reveals a drastic clustering of COs at the distal ends of $axr1^{-/-}$ chromosomes

To obtain an accurate description of CO distribution along the chromosomes in both $axr1^{-/-}$ and wild-type backgrounds, we compared the male genetic maps of the five chromosomes. We backrossed F1 hybrids (Col-0 x Ws, wild type or $axr1^{-/-}$) as male to Col-0 as female. Genetic maps were obtained through segregation analysis of 96 SNP markers in the progeny (as described in [38]. The mean number of observed breakpoints per plant in the offspring was the same in wild type (5.1 (95% CI: 4.78–5.34)) and $axr1^{-/-}$ (5.2 (95% CI: 5.02–5.47)) (Fig 1A). This is in agreement with our previous cytological survey that showed that the number of class I COs was unchanged in $axr1^{-/-}$ [37]. At the chromosome scale, we also did not detect any significant difference in the number of chromatids with 0, 1, 2 or more COs in $axr1^{-/-}$ compared to wild type (S1 Fig).

However, CO distribution along the chromosomes was dramatically altered in the mutant (Fig 1B and 1C; S1 Table). For each of the five chromosomes, a large central region of the chromosomes was almost devoid of COs, whereas CO rates were considerably higher in terminal regions. For example, the 2.6 Megabases (Mb) subtelomeric region constituting 8.7% of the physical length of chromosome 1 was 34 centiMorgan (cM) (27% of the genetic length) in $axr1^{-/-}$ compared to 11 cM (8.8% of the genetic length) in the wild type (Fig 1B). Conversely, the 9.1 Mb centromere proximal region on chromosome 1 (30% of the physical length) was 39.5 cM in the wild type and more than four times shorter, at only 8.6 cM, in $axr1^{-/-}$ (7% of the genetic length) (Fig 1B). The CO rate was significantly different in 18 genetic intervals (Benjamini, Krieger and Yekutieli, with Q = 1%) between the wild type and the $axr1^{-/-}$ mutant: four intervals were either immediately adjacent or very close to telomeres with a large increase in CO rates in the mutant whereas the others were proximal with lower CO rates in $axr1^{-/-}$ compared to wild type (Fig 1B, S1 Table). When the $axr1^{-/-}$ CO rates in each interval were plotted against the relative distance of the interval from the centromeres, $axr1^{-/-}$ CO rates increased exponentially ($r^2$ = 0.8745) towards the telomeres and were much higher than wild-type CO

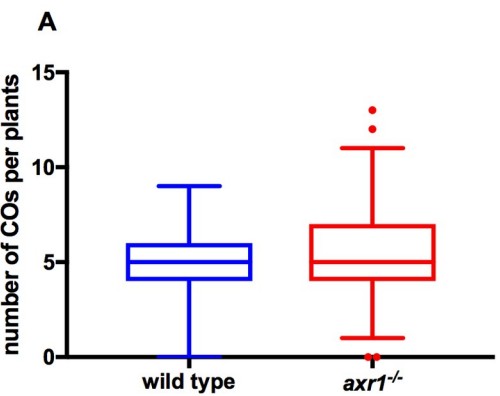

**Fig 1. Characterization of the number and the localization of meiotic crossovers. A:** number of COs per plants in wild type and *axr1*⁻/⁻. Box plots with 1 and 99 percentile (number of wild-type plants = 173; number of *axr1*⁻/⁻ plants = 353) **B:** CO rates during male meiosis along the five chromosomes. Intervals with significantly different CO rates between wild type (blue curve) and *axr1*⁻/⁻ (red curve) are indicated with black stars (multiple t tests with the procedure Benjamini, Krieger and Yekutieli with Q = 1%). Arrows on the x-axis of chromosome 3 and chromosome 4 indicate the localization of the BACs chosen for FISH analysis. Grey boxes on chromosome 1 indicate the intervals used to compare the difference in genetic length between wild type and *axr1*⁻/⁻. Black squares on the X axes: centromere position . Error bars: 95 % confidence intervals (CI). **C:** CO rates from centromeres to telomeres. The x-axis is the relative distance from the telomeres. The curves were obtained by interpolating a non-linear regression analysis using Prism and Graphad Software. Blue squares and blue line: wild type. Red circles and red line: *axr1*⁻/⁻ **D:** distances between two COs on the same chromatid. Distances are expressed in percentage of chromosome length. Chromatids with two and only two COs were selected. n = 181 in wild type; n = 336 in *axr1*⁻/⁻.

rates. In contrast, CO rates were always lower than wild-type rates in the proximal third of the chromosomes (Fig 1C).

We also observed that the physical distances between two COs occurring on the same chromatid were drastically shorter in *axr1*⁻/⁻ than in wild type (Fig 1D) (t-test two-tailed p-value $< 10^{-8}$). In wild-type plants, COs are, on average, separated by half a chromosome (50%, 9 to 15 Mb) and they are rarely close to each other or at the two ends of the chromatids (Fig 1D). In contrast, in *axr1*⁻/⁻, when two COs were observed on the same chromatid, they were either close to each other (up to 20% of the chromosome, 4 to 6 Mb in average) or one at each distal end of the chromosome (Fig 1D). The distributions of the physical distances between two COs differ radically between *axr1*⁻/⁻ and the wild type and they also differ from a random distribution (Fig 1D). In summary, impairment of the neddylation pathway leads to a redistribution of COs with clustering at the distal ends of chromosomes.

## The telomere bouquet forms normally in *axr1*⁻/⁻ male meiocytes.

We next investigated whether CO redistribution at chromosomal distal regions in *axr1* mutants could be correlated to either the formation or dynamics of telomere clustering at the nuclear envelope, a process known as the telomere bouquet. The telomere bouquet has been illustrated in several plant, fungi and animal species [39,40]. To investigate bouquet behavior, we developed a method that preserves the 3D-configuration of nuclei and allows the labeling of DNA or proteins by immunostaining or/and FISH [41]. Using this technique in the mutant during early prophase I, one third of the meiocytes displayed a clear telomere bouquet conformation [41] (Fig 2A and 2B). At a given time point, not all the telomeres were included in the bouquet with a proportion ranging from 35 to 80% (Fig 2C). Similar results were observed in the wild type [41] suggesting that bouquet dynamics are not modified in the mutant. We then compared the volume occupied by the bouquet between the mutant and wild type. In the wild type, the mean volume of the telomere bouquet was approximately 44.7 μm³ (± 8.2) [41]. In *axr1*⁻/⁻ meiocytes, the volume was not significantly different (51.4 μm³ ± 10.3) (p = 0.43) (Fig 2D). We thus conclude that bouquet organization is not modified in *axr1*⁻/⁻.

## Chromosome pairing is defective in *axr1*⁻/⁻

We then examined chromosome behavior during prophase of the first meiotic division in *axr1*⁻/⁻ compared to wild type. We performed FISH on intact meiocytes with bacterial artificial chromosome (BAC) probes to study the pairing behavior of the corresponding regions during meiotic prophase. Chromosomal regions were considered paired when only one foci was visible or when two foci were visible and separated by less than 0.5 μm. We chose BACs that target the distal and proximal regions, on chromosome 3 or chromosome 4, strongly affected in terms of CO rates between *axr1*⁻/⁻ and wild type (arrows on Fig 1B). The two targeted proximal regions did not recombine or recombined at only very low levels in the mutant whereas in the same regions in the wild-type CO rates were above or close to the chromosome average

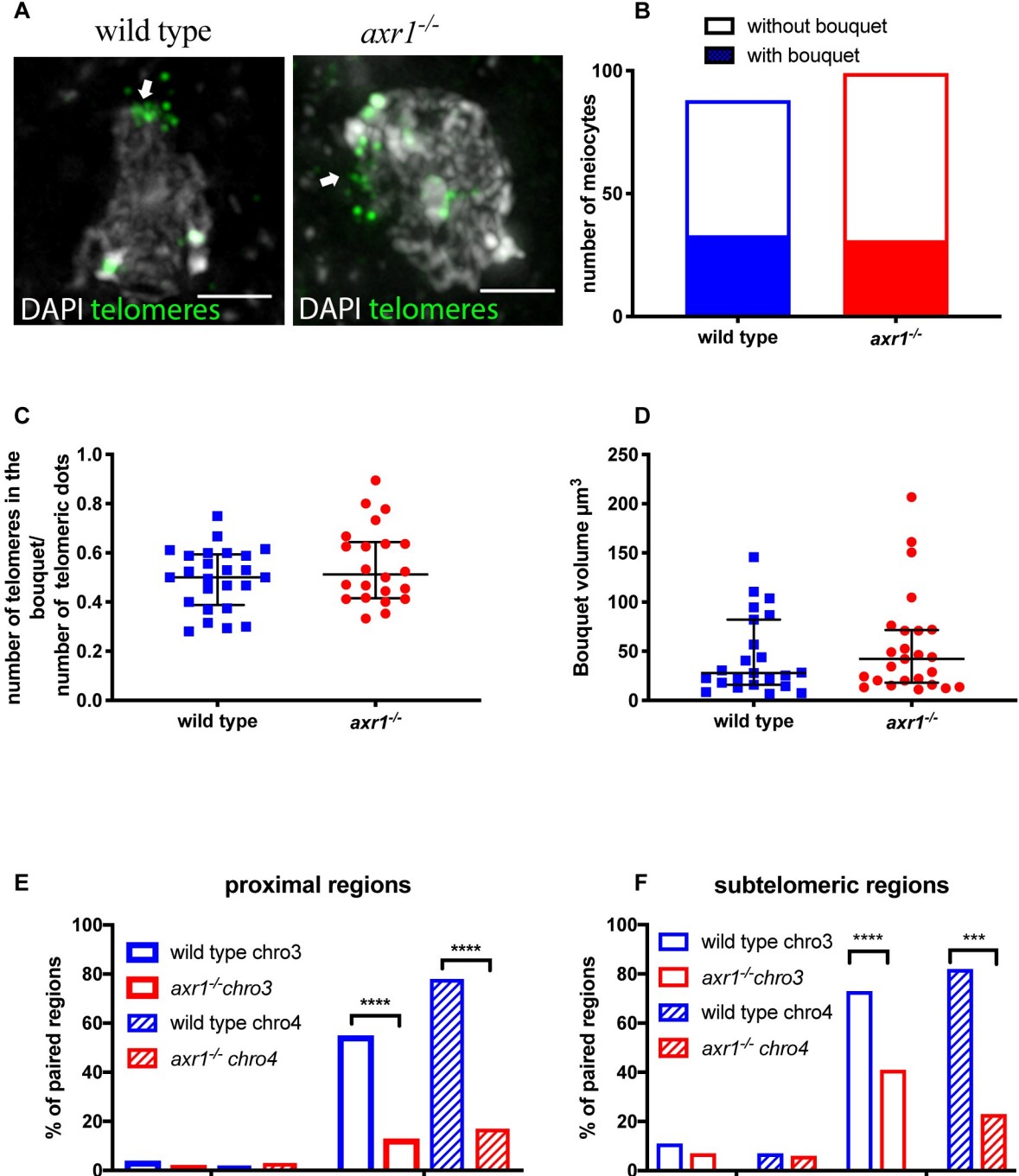

**Fig 2. Dynamic of chromosomes during meiosis prohase: telomeres localization and pairing. A:** FISH on intact wild type and *axr1*⁻/⁻ meiocytes with the telomeric probe pTat4. The pictures correspond to projections of z-sections obtained by epifluorescence microscopy followed by deconvolution. In Prophase I telomeres cluster to one side of the nucleus (white arrow). Scale bar 2μm **B:** Quantification of the proportion of meiocytes displaying a telomere bouquet. (wild type n = 107; *axr1*⁻/⁻ n = 102) **C:** Quantification of the proportion of telomeres included in the bouquet (wild type n = 24; *axr1*⁻/⁻ n = 22). **D:** Bouquet volume in μm³. The volume was calculated as described in the Material and Methods. (wild type n = 23, *axr1*⁻/⁻ n = 25). Wild type (wt) = Col-0; Red circles *axr1*⁻/⁻ = SAIL_904_E06. Black bars represent the median with the interquartile range. **E:** Quantification of the pairing of proximal regions of chromosome 3 (empty rectangles) and chromosome 4 (striped rectangles) in early meiosis/G2 and Prophase I (wild type (blue)) chromosome 3 early/G2 n = 46 prophase I n = 175; chromosome 4 early/G2 n = 46 prophase I n = 138; *axr1*⁻/⁻ (red) chromosome 3 early/G2 n = 53 prophase I n = 53; chromosome 4 early/G2 n = 53 prophase I n = 53).(Prophase I,

chromosome 3 chi-square test, two-sided, p-value = 3.7 $10^{-8}$; chromosome 4 chi-square test, two-sided, p-value<$10-10$) **F:** Quantification of the pairing of subtelomeric regions of chromosome 3 (empty rectangles) and chromosome 4 (striped rectangles) in early meiosis/G2 and Prophase I. Wild type chromosome (blue): early/G2 n = 47 prophase I n = 132; chromosome 4 early/G2 n = 29 prophase I n = 111. $axr1^{-/-}$ (red) chromosome 3: early/G2 n = 43 prophase I n = 51; chromosome 4 early/G2 n = 35 prophase I n = 43.(Prophase I, chromosome 3 chi-square test, two-sided, p-value = 6.7 $10^{-5}$; chromosome 4 chi-square test, two-sided, p-value = 2.9 $10^{-4}$). Wild type (wt) = Col-0; $axr1^{-/-}$ = SAIL_904_E06.

(arrows on Fig 1B). In contrast, the two subtelomeric regions showed a 2.5 to five fold increase in CO rates in the mutant background compared to wild type. In the premeiotic G2 phase, there was no significant pairing of the two proximal regions either in wild type or $axr1^{-/-}$ (Fig 2E; S2 Fig) and very little pairing for subtelomeric regions in both genetic backgrounds (Fig 2F; S2 Fig). Later in prophase I in the wild type, both proximal and distal regions on both chromosomes were paired in a large majority of the male meiocytes (from 56% to 82% depending on the chromosomal region, 72% on average) (Fig 2E and 2F; S2 Fig). In the mutant, pairing was also detected for each of the four chromosomal regions but at a significantly lower level (from 17% to 41% of the cells, average 23% of the cells) (Fig 2E and 2F; S2 Fig). This deficit in pairing was observed for both proximal and distal regions. The impact on pairing was, however, slightly less pronounced on distal (around four times less) versus proximal regions (two to three times less) (Fig 2E and 2F; S2 Fig). In conclusion, homologous chromosome pairing is dramatically reduced in both proximal and distal regions in $axr1^{-/-}$ despite CO rates are increased in distal regions compared to wild type.

## No differences in early recombination events are detected between wild type and $axr1^{-/-}$

We previously analysed the appearance of DMC1 (a meiosis-specific protein that marks recombination sites) on spread male meiocytes, showing that the average number of DMC1 foci was not different from wild type in $axr1^{-/-}$ [37]. To confirm this result and to exclude that changes in CO location observed in $axr1^{-/-}$ were missed due to the 2D spread technique, we immuno-localised DMC1 on 3D-structure preserved meiocytes [41]. We co-immunolocalised DMC1 with ASY1, a major component of the meiotic chromosome axis, and H3K9me2, a chromatin mark enriched in the heterochromatin. We found that the timing of appearance and the global distribution of DMC1 was unchanged in $axr1^{-/-}$. In wild type as in $axr1^{-/-}$, DMC1 foci are detected associated with the axis or the chromatin from leptotene to zygotene, in all regions of the chromosomes but heterochromatic ones (S3 Fig).

## Methylated DNA regions are organized differently in wild-type and $axr1^{-/-}$ male meiocytes

As DNA methylation is known to affect CO distribution, we evaluated the possibility that CO relocalization in $axr1^{-/-}$ is caused by methylation changes through immunolabeling 5-methyl-cytosine (5mC) in male meiotic cells. As described previously [42], we observed that 5mC staining was restricted to pericentromeric regions in wild type (Fig 3A). In $axr1^{-/-}$ male meiocytes, the 5mC stretches were twice as long as those of wild type, suggesting an extension of DNA methylation (t-test two-tailed p-value $< 10^{-10}$) (Fig 3B). However, both pericentromeric dense region and the Nucleolar Organizer Region (NOR) measured by DAPI appear to be unmodified (S4 Fig). Thus, large central DNA hypermethylated chromosomal regions are observed in $axr1^{-/-}$ meiocytes and the methylated regions detected in $axr1^{-/-}$ do extend further than only the dense DAPI-stained heterochromatic regions, suggesting a different organization of the chromatin in $axr1^{-/-}$ male meiocytes.

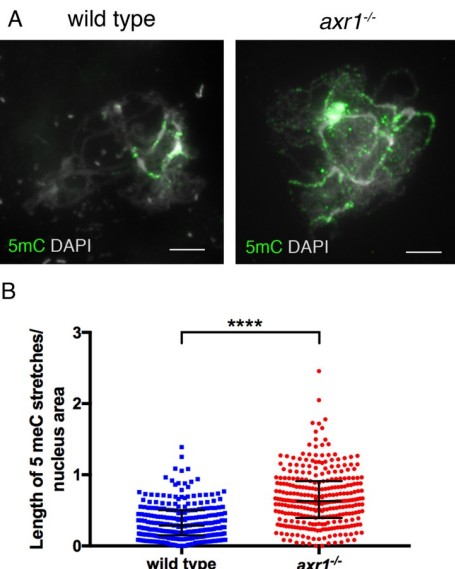

**Fig 3. Characterization of 5-methyl-cytosine in meiocytes from wild-type and *axr1*<sup>-/-</sup>. Plants A:**
Immunolocalization of 5-methyl-cytosine (5-mC) in prophase 1 zygotene using an antibody directed against 5-mC. Scale bar = 5μm **B:** Quantification of the labelling of 5-mC stretches. To compare nuclei accurately, for each nucleus the length of the immunolabelled stretches was divided by the nucleus area (wild type n = 32; *axr1*<sup>-/-</sup> n = 30). Black bars represent the median with interquartile range. Unpaired t-test two-tailed p-value $<10^{-10}$. (wt = Col-0; *axr1*<sup>-/-</sup> = SAIL_904_E06).

## Transposable elements are hypermethylated in *axr1*<sup>-/-</sup>

To better describe the extent of the changes observed in immunocytology assays, we extracted wild type and *axr1*<sup>-/-</sup> male meiocytes and performed bisulfite sequencing using a protocol described previously [31,43]. The levels of methylation per cytosine were determined and confirmed that biological replicates clustered together (S5A Fig). DNA methylation at chromosomal level was similar in both the CG and CHG contexts between the wild type and *axr1*<sup>-/-</sup> male meiocytes, but was increased in *axr1*<sup>-/-</sup> for CHH in the centromeric and pericentromeric regions enriched in TEs (Fig 4A, S5B Fig). Consistently, the methylation over genes was similar in *axr1*<sup>-/-</sup> and wild type meiocytes in all sequence contexts, whereas for TEs, CHH methylation was notably increased in *axr1*<sup>-/-</sup> (Fig 4B, S5C Fig). Thus, in meiocytes, we observed a specific increase for CHH methylation in *axr1*<sup>-/-</sup> pericentromeric regions and TEs.

We then sought to understand if the effect of *axr1*<sup>-/-</sup> on DNA methylation is specific to the meiocytes, or general among plant tissues. Genomic DNA was extracted from leaves at similar stages in *axr1*<sup>-/-</sup> and wild-type plants and sequenced after bisulfite conversion. Clustering analyses, based on CG methylation, revealed that all three replicates were similar for each genotypes (S6A Fig). The genome was divided in non-overlapping 200 kb-tiles and their average methylation levels were calculated revealing a global increase in DNA methylation in *axr1*<sup>-/-</sup> somatic cells, more prominently in pericentromeric regions like in meiocytes (Fig 5A). On average, gene methylation was similar in *axr1*<sup>-/-</sup> and wild-type somatic cells (Fig 5B; S6B Fig). TEs, on the other hand, were hypermethylated in all contexts in *axr1*<sup>-/-</sup> leaves (Fig 5B; S6B, S6C and S7 Figs). CG, CHG and CHH methylation levels of *axr1*<sup>-/-</sup> TEs were increased by 15%, 53% and 39%, respectively, compared to wild-type TEs (Fig 5B). In addition, we found that this hypermethylation was widespread and not restricted to a specific family of TEs or

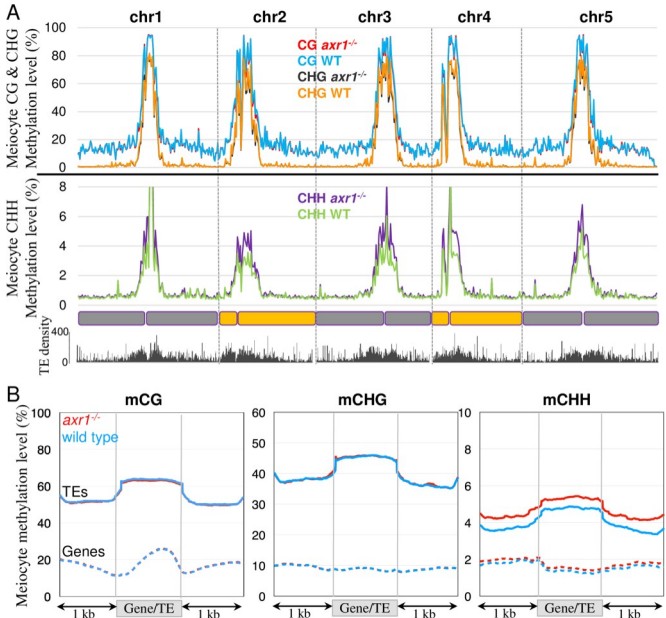

**Fig 4. Patterns of DNA methylation in male meiocytes. A:** Methylation along chromosomes, calculated from non-overlapping 200 kb bins. The average methylation levels were determined by combining the biological replicates for each genotype. **B:** Patterns of methylation in genes and TEs in the Col-0 wild type (blue lines) and the *axr1⁻/⁻* mutant (red lines). The average methylation level of genes and TEs was determined by dividing the corresponding annotated regions into 100 bp bins. Regions located 1 kb upstream and 1 kb downstream of the gene bodies or TEs are shown. The average methylation levels were determined by combining the biological replicates for each genotype.

repeats (S8 Fig). Sequencing the leaf methylome of another *axr1* allele (i.e. *axr1-12* [37]) confirmed the results (S6B Fig).

The regions that were significantly differentially methylated (DMRs) between *axr1⁻/⁻* and the wild type were identified. The somatic cells of *axr1⁻/⁻* contained a high number of hyperDMRs in the CHG context (n = 13,708) and to a lesser extent in the CG and CHH contexts (n = 462 and 970, respectively; Fig 5C and S3 Table). HyperDMRs mainly overlapped with TEs (44% of the CG hyperDMRs, 77% of the CHG hyperDMRs and 76% of the CHH hyperDMRs) that were already methylated in the wild type (S9 Fig) and are mostly localised within the pericentromeric regions (S10 Fig). In contrast, we identified a much more limited number of hypoDMRs (n = 70 for CGs, 2 for CHGs and 2 for CHHs; S4 Table). Therefore, DMR analysis showed that TEs in the *axr1⁻/⁻* somatic cells are hypermethylated in all cytosine contexts.

We have previously shown that CG and CHG methylation are strongly reinforced in the germ cells (including the meiocyte) in comparison to somatic tissues [31,44,45]. Indeed, CG and CHG methylation in wild-type (and *axr1⁻/⁻*) meiocytes mimic those in *axr1⁻/⁻* leaves (Figs 4B and 5B). In other words, although *axr1⁻/⁻* mutation increases CG and CHG methylation in soma (Fig 5B), it cannot further increase CG and CHG methylation in meiocyte, whose methylation levels are already as high as *axr1⁻/⁻* leaves (Figs 4B and 5B). This result suggests that *axr1⁻/⁻* mutation affects general methylation in the plant, rather than specfically affecting the methylation in meiocytes. Consistent with this idea, in the CHH context, where meiocytes have lower CHH methylation compared to leaves (Figs 4B and 5B) [31], *axr1⁻/⁻* mutation induces CHH hypermethylation in meiocytes (Fig 4B). Taken together, our results show that the effect of *axr1⁻/⁻* mutation on meiocyte methylome occurs prior to reproductive development, by affecting general DNA methylation pathways.

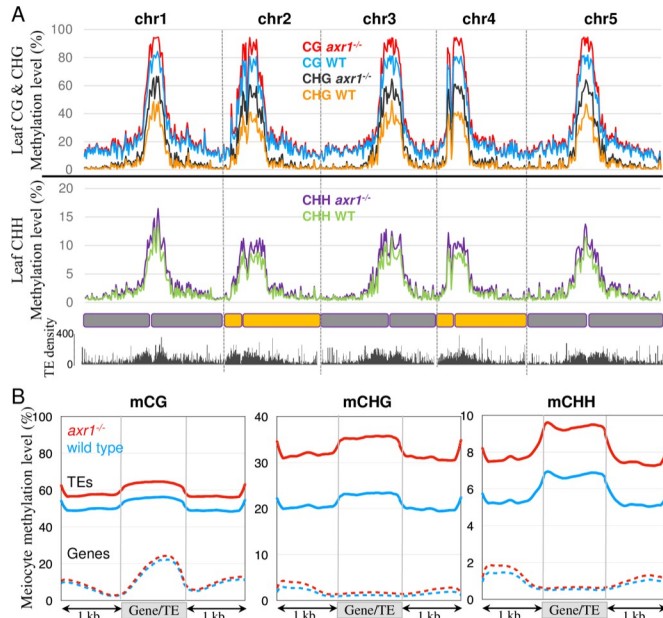

**Fig 5. Patterns of DNA methylation in somatic cells. A:** Methylation along chromosomes, calculated from non-overlapping 200 kb bins. The average methylation levels were determined by combining the biological replicates for each genotype. **B:** Patterns of methylation in genes and TEs in the Col-0 wild type (blue lines) and the $axr1^{-/-}$ mutant (red lines). The average methylation level of genes and TEs was determined by dividing the corresponding annotated regions into 100 bp bins. Regions located 1 kb upstream and 1 kb downstream of the gene bodies or TEs are shown. The average methylation levels were determined by combining the biological replicates for each genotype.

## Impairment of DNA methylation pathways is epistatic over $axr1^{-/-}$ for DNA methylation in somatic cells

To get further insight into the molecular pathways involved in the hypermethylation of TEs, we examined whether the hyperDMRs detected in $axr1^{-/-}$ somatic cells overlapped with DMRs found between mutants impaired in DNA methylation homeostasis and their respective wild-type controls. 85% (n = 395) of $axr1^{-/-}$ CG hyperDMRs overlap with $met1^{-/-}$ CG hypoDMRs, suggesting that most of the CG sites hypermethylated in $axr1^{-/-}$ are maintained by MET1 in the wild type. 88% (n = 12,097) of $axr1^{-/-}$ CHG hyperDMRs overlap with $cmt3^{-/-}$ CHG hypoDMRs and 55% (n = 7,601) with $kyp^{-/-}$ CHG hypoDMRs. Thus, the majority of $axr1^{-/-}$ CHG hyperDMRs matches regions controlled by the CMT3/KYP pathway. Finally, 68% (n = 661) of $axr1^{-/-}$ CHH hyperDMRs overlap with $pol\ iv^{-/-}$ CHH hypoDMRs and 24% with $cmt2^{-/-}$ CHH hypoDMRs, suggesting that $axr1^{-/-}$ CHH hyperDMRs are mainly targeted by the RdDM canonical pathway in the wild type. In addition, $axr1^{-/-}$ CG- CHG- and CHH hyperDMRs poorly overlap between them (1% of the 13,708 CHG hyperDMRs overlap with CG hyperDMRs and 2% with CHH hyperDMRs). Therefore, methylation pathways are ubiquitously impaired in $axr1^{-/-}$, suggesting that a more general pathway acting upstream of DNA methylation is likely compromised in the mutant.

$axr1^{-/+}$ heterozygous plants were crossed with mutants individually impaired for different methylation pathways, namely $nrpd1a-4^{-/-}$ ($pol\ iv$), $cmt3-11^{-/-}$ and $met1-1^{-/-}$. DNAs extracted from leaves (bulks from 10 plants) of $axr1^{-/-}\ met1^{-/-}$, $axr1^{-/-}\ pol\ iv^{-/-}$ and $axr1^{-/-}\ cmt3^{-/-}$ F2 progenies were sequenced after bisulfite conversion. Leaves from the corresponding single $met1^{-/-}$ or $cmt3^{-/-}$ or $pol\ iv^{-/-}$ mutant siblings were used as controls. We first examined the CG methylation controlled genome-wide by MET1. By comparing the methylomes of $axr1^{-/-}\ met1^{-/-}$ and

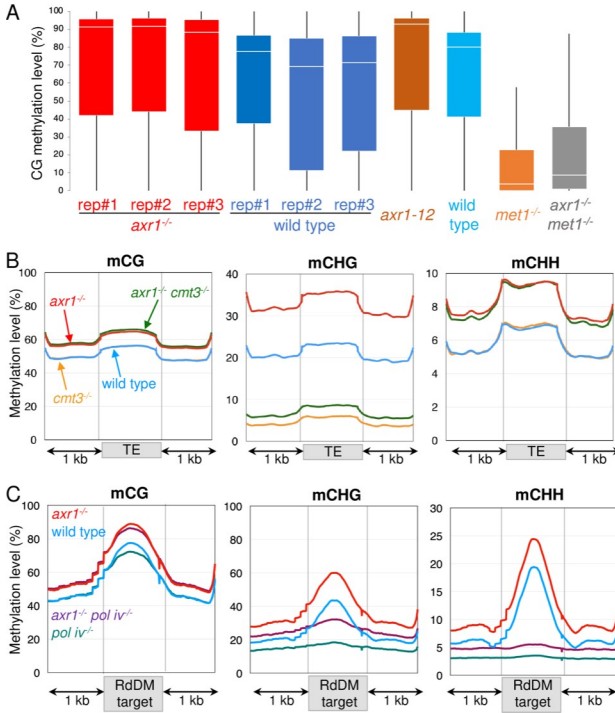

**Fig 6. Patterns of DNA methylation in somatic cells of *axr1*[-/-] *cmt3*[-/-], *axr1*[-/-] *met1*[-/-] and *axr1*[-/-] *pol iv*[-/-] double mutants. A:** Boxplots showing mean methylation content of *axr1*[-/-] or *axr1-12* mutants and the corresponding Col-0 wild type (WT) compared to *axr1*[-/-] *met1*[-/-] and *met1*[-/-] mutants. The *Arabidopsis* genome (TAIR10) was partitioned in non-overlapping 100 bp-tiles and methylation levels correspond to the ratios of methylated cytosines over the total number of cytosines. Only cytosines covered by at least five reads were considered. All biological replicates are shown. **B and C:** Patterns of methylation in TEs (**B**) and RdDM targets (**C**) in the Col-0 wild type, *axr1*[-/-], *cmt3*[-/-], *pol iv*[-/-] single mutant and the double *axr1*[-/-] *cmt3*[-/-] and *axr1*[-/-] *pol iv*[-/-]. The average methylation level of TEs or RdDM targets was determined by dividing the corresponding annotated regions into 100 bp bins. Regions located 1 kb upstream and 1 kb downstream are shown. The average methylation levels were determined by combining the biological replicates for each genotype.

*met1*[-/-] F2 plants, a limited number of CG DMRs was identified (0 hypoDMR and 123 hyperDMRs), indicating that the CG methylation patterns of *axr1*[-/-] *met1*[-/-] and *met1*[-/-] somatic cells are similar. This result was confirmed by determining the average levels of CG methylation of non-overlaping 100-bp bins spanning the whole genome (Fig 6A). Similar results were obtained for the CHG maintaining pathway when the methylomes of *axr1*[-/-] *cmt3*[-/-] and *cmt3*[-/-] F2 plants were compared since a limited number of CHG DMRs was identified between the mutants (0 hypoDMR and 40 hyperDMRs), indicating that the CHG methylation patterns of *axr1*[-/-] *cmt3*[-/-] and *cmt3*[-/-] somatic cells are similar. The general profiles of methylation for TEs that are targeted by the CMT3/KYP pathway confirmed the results. On average, TEs were similarly CHG hypomethylated in *axr1*[-/-] *cmt3*[-/-] and *cmt3*[-/-] leaves, while CG and CHH hypermethylation patterns were identical in both *axr1*[-/-] *cmt3*[-/-] and *axr1*[-/-] (Fig 6B). The same results were obtained for the canonical RdDM pathway when we compared the methylomes of *axr1*[-/-] *pol iv*[-/-] and *pol iv*[-/-] mutant leaves. We found only 146 CHH hyperDMRs between *axr1*[-/-] *pol iv*[-/-] and *pol iv*[-/-] and 5 CHH hypoDMRs. The patterns of CHH hypomethylation for TEs specifically targeted by the RdDM pathway in the wild type, were also identical between *axr1*[-/-] *pol iv*[-/-] and *pol iv*[-/-] mutants (Fig 6C). Altogether, the results indicate that mutations affecting DNA methylation are epistatic over *axr1*[-/-] in somatic cells, although we note that a few amount of DMRs found in the *axr1*[-/-] background are still

detected in the double $axr1^{-/-}$ $met1^{-/-}$ (0.6% of the $axr1^{-/-}$ CG hyperDMR), $axr1^{-/-}$ $cmt3^{-/-}$ (7% of the $axr1^{-/-}$ CHG hyperDMR) and $axr1^{-/-}$ $pol\ iv^{-/-}$ (1.6% of the $axr1^{-/-}$ CHH hyperDMR).

## Impairment of DNA methylation does not restore regular meiosis in $axr1^{-/-}$

To understand whether the $pol\ iv^{-/-}$, $cmt3^{-/-}$ or $met1\text{-}1^{-/-}$ mutations were also epistatic over $axr1^{-/-}$ in meiosis, we analysed chromosome segregation in male meiocytes of these single mutants, or in combination with $axr1^{-/-}$. In $cmt3^{-/-}$ and $pol\ iv^{-/-}$ single mutants, chromosome segregation was regular with always five bivalents formed at metaphase I as in wild type (S11 Fig). Previous reports indicate that meiotic chromosome segregation and morphology of $met1^{-/-}$ mutants are similar to the wild type [33]. However, in the double $axr1^{-/-}$ $cmt3^{-/-}$, $axr1^{-/-}$ $pol\ iv^{-/-}$ and $axr1^{-/-}$ $met1^{-/-}$ mutants, meiosis ressembles $axr1^{-/-}$ meiosis with an absence of fully synapsed chromosomes (pachytene stage) and a comparable shortage of bivalents (Fig 7A– 7E). We quantified the localization of the remaining chiasmata based on bivalent configuration as described in [46]. In all double mutants as in the single $axr1^{-/-}$ mutant, around two third of the chiasmata were distal (Fig 7F). In both the wild type and the single mutants, the chiasmata

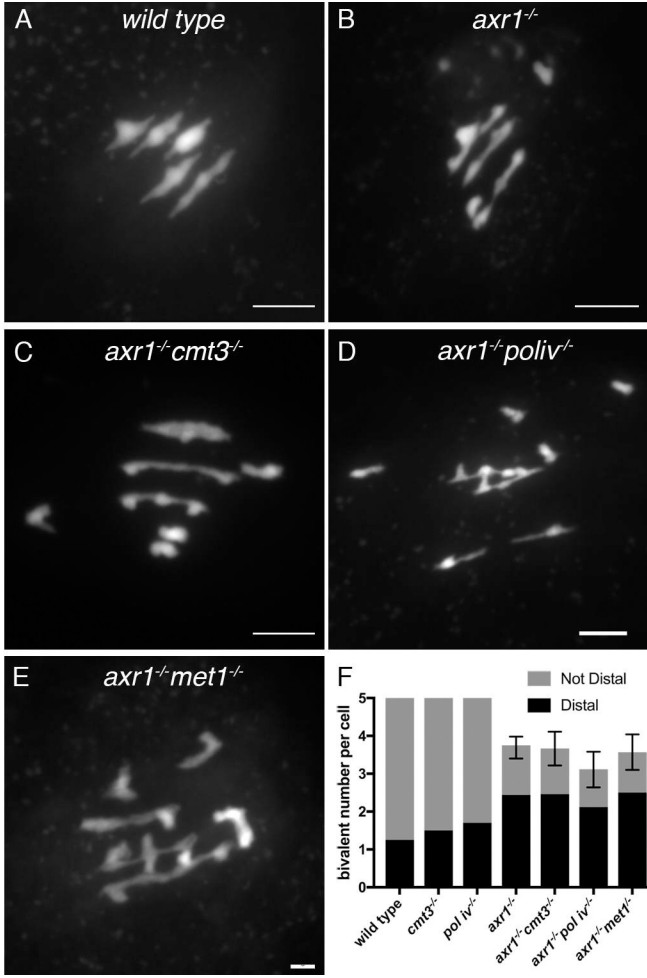

**Fig 7. Chromosome configuration at metaphase 1. A-E:** Representative examples of metaphase 1 cells from wild type (**A**), $axr1^{-/-}$ (**B**), the double $axr1^{-/-}$ $cmt3^{-/-}$ (**C**) $axr1^{-/-}$ $pol\ iv^{-/-}$ (**E**) and $axr1^{-/-}$ $met1^{-/-}$ (**F**). **G**: mean number of bivalents per cells with the chiasma localization (Error bars: Upper and Lower 95% Confidence Intervals).

were mostly intersticial (Fig 7F). Thus, mutations affecting methylation do not restore bivalent formation in *axr1*[-/-] meiosis. Altogether, our data show that the redistribution of COs observed in *axr1*[-/-] can be uncoupled from TE hypermethylation changes by mutating either *MET1* or *CMT3* or *Pol IV*.

## Discussion

Using genome-wide molecular genetic approaches, we confirmed that in *axr1*[-/-], meiotic COs are mislocalized despite similar CO rates in wild type and mutant. We extended these observations by showing that (i) chromosome pairing is defective in *axr1*[-/-] meiocytes, (ii) CO events are redistributed towards distal chromosomic regions, (iii) *axr1*[-/-] TEs are hypermethylated in all contexts for somatic cells and in the CHH context for meiocytes, (iv) mutations affecting methylation maintenance are epistatic over *axr1*[-/-] in terms of somatic DNA methylation whereas, for meiotic CO formation, *axr1*[-/-] is epistatic to mutations affecting methylation maintenance, thus uncoupling DNA methylation from the formation of COs in the *axr1*[-/-] mutant background.

In *axr1*[-/-], impairment of the neddylation pathway confers a distinctive CO patterning with clusters of COs localizing to the distal parts of the chromosomes (Fig 1B and 1C). Indeed, the competency of the distal regions for CO formation appears to be exacerbated as CO rates increase exponentially from centromeres to telomeres in the mutant (Fig 1C). CO patterning depends on a series of interlinked controls acting on the meiotic recombination machinery. The first layer acts at the recombination initiation step. However, this step is not modified in *axr1*[-/-] because the number, distribution, and dynamics of the DMC1 protein, a meiosis specific recombinase that forms foci at DSB sites, has not revealed major differences in *axr1*[-/-] compared to wild type (S3 Fig). Thus, the CO formation control is likely impaired at a subsequent stage in *axr1*[-/-]. In *Arabidopsis* as in many species, DSB repair, occurring during early prophase, mediates pairing between homologous chromosomes which then develops during the zygotene stage into a closely and structured association called synapsis. The global pairing defect observed in *axr1*[-/-] (Fig 2E and 2F) suggests that the DSB repair step on the intact homologous chromosome is somehow impaired or that it does not provide a robust interaction between the homologous chromosomes over their entire length. However, some of these interactions are competent for CO formation leading to HEI10 foci and/or synapsis progression [37]. Indeed, we previously reported that partial synapsis occurs but overall synapsis was strongly affected in *axr1*[-/-]. Localized chiasma in restricted regions of some chromosomes that undergo pairing were also reported in various fungi, animals and plants [47,48] but these were rarely associated with CO clustering. Only the *Arabidopsis pss1* and the *Sordaria mer2-17* show such a phenotype, but to a lesser extent and with a less pronounced synapsis defect than in *axr1*[-/-] [28,49]. In *mer2-17*, the pairing defect observed was suggested to reduce the number of DSBs competent for CO formation leading to clusters of Hei10 foci in portions of synapsed chromosomes [49]. Thus, there could be a mechanism controlling the total CO number per meiocyte that would force CO designation in small portions of paired chromosomes leading to the observed CO clustering in synapsed regions. According to this hypothesis, in *axr1*[-/-], despite the global pairing disorder, there would be still a signal present which mediates the formation of a "wild type number" of CO precursors in some restricted regions. The distal regions of the chromosomes appear to be much more competent for CO formation than the other chromosomal regions. This observation is in apparent contradiction with (i) the fact that the pairing efficiency of the terminal regions was reduced by up to 3-fold in the mutant compared to the wild type and (ii) that in average per meiosis one pair of chromosomes does not form a CO. Nevertheless, pairing is less affected in distal regions than in proximal regions suggesting that the distal regions are more prone to form clusters of COs.

The distal expansion of the genetic map in $axr1^{-/-}$ is correlated with a dramatic map contraction in large proximal regions surrounding centromeres. In the mutant, the regions with low CO rates represent 30 to 40% of the chromosome arm length. This largely exceeds the pericentromeres as defined in the wild type (heterochromatic regions enriched in methylated TEs and repeated sequences) that represent only around 10% of the chromosome arm length. In *Arabidopsis*, several reports indicate that DNA methylation interacts with meiotic recombination at various levels. First, when CG and/or non-CG methylation is reduced, meiotic DSBs are detected in centromeric regions [16,32], suggesting that DNA methylation limits DSB formation in these regions. Loss of CG methylation in *Arabidopsis* increases CO formation in chromosome arms [33–36] whereas CO number increases slightly in pericentromeric regions when the pathways maintaining non-CG methylation are compromised [32]. However, these hypotheses are all based on data obtained with methylomes of somatic cells and could probably be redefined by sequencing methylomes from meiocytes. Indeed, we already reported that patterns of methylation between meiocytes and somatic cells are different (31), a result that we confirm in the present study by sequencing $axr1^{-/-}$ meiocytes. Whereas a dramatic cytosine hypermethylation was observed for TEs in all contexts (CG, CHG, CHH) of $axr1^{-/-}$ somatic cells (Fig 5B), the methylomes of meiocytes show that TEs of $axr1^{-/-}$ are only hypermethylated in the CHH context compared to wild type (Fig 4). Consistently, we report that methylated regions detected by immunocytology do extend further in $axr1^{-/-}$ than in wild type meiocytes (Fig 3). We believe the reason why CG and CHG methylation are not increased in $axr1^{-/-}$ meiocytes like they do in the somatic tissue, is that CG and CHG methylation are already strongly reinforced in the meiocyte, as described in our previous study [31]. Indeed, CG and CHG methylation in wild type (and $axr1^{-/-}$) meiocytes mimic those in $axr1^{-/-}$ leaves (Figs 4B and 5B). The comparison between meiocyte and leaf methylomes suggests that $axr1^{-/-}$ affects general methylation in the plant, rather than specficially affecting the methylation in meiocytes. Nevertheless, when the pathways maintaining the methylation homeostasis are compomised, regular meiotic chromosome segregation is not restored in $axr1^{-/-}$ whereas cytosine methylation is effectively impaired (Fig 6). Thus, in the meiocytes of $axr1^{-/-}$, cytosine methylation *per se* does not seem to be the main factor responsible for the redistribution of COs along the chromosomes, although the methylome of $axr1^{-/-}$ $pol\ iv^{-/-}$ meiocytes remain to be sequenced to confirm that CHH hypermethylation is not involved.

The *AXR1* gene codes for the E1 enzyme of the neddylation pathway involved in the activation of the cullin ring ligases (CRLs). CRLs are central to numerous processes including hormones perception, response to stresses, regulation of cell-cycle and transcription [50]. There are three main cullin types in *Arabidopsis*, namely CUL1-CUL2, CUL3a-b, and CUL4 [50]. We previously showed that the deregulation of CO localization in *Arabidopsis* $axr1^{-/-}$ mutant is likely to be mediated by the CRL4 complex [37]. Whether the same CRL complex also limits the methylation of TEs will have to be further investigated. Mutating *AXR1* compromises a pathway involved in the methylation setup but likely localised upstream of DNA methyltransferases. This chromatin-specific pathway, crucial to permit proper DNA methylation of TEs, would remain impaired in the double $axr1^{-/-}$ $met1^{-/-}$, $axr1^{-/-}$ $pol\ iv^{-/-}$ or $axr1^{-/-}$ $cmt3^{-/-}$ mutant backgrounds as in $axr1^{-/-}$. Such pathway could be related to chromatin remodelers or histones specific of heterochromatin such as histone H1, which binds linker DNA between two adjacent nucleosomes in heterochromatin [51,52], or DECREASE IN DNA METHYLATION 1 (DDM1), a nucleosome remodeler that facilitates methylation of heterochromatic TEs in H1-enriched regions [53,54].

In conclusion, our data demonstrate that plant neddylation pathways play crucial roles in both controling the localization of meiotic COs, without affecting their formation rate, and in restricting the methylation of TEs by a pathway that remains to be more specifically characterized.

## Materials and methods

### Plant materials and growth conditions

The *Arabidopsis* Col-0 lines, including *axr1* mutant lines (SAIL_904_E06 = *axr1*[-/-]; N3076 = *axr1-12*), were obtained from the Salk Institute Genomic Analysis Laboratory (SIGnAL, http://signal.salk.edu/cgi-bin/tdnaexpress) and provided by NASC (http://nasc.nott.ac.uk/). The *Arabidopsis* Ws-4 lines including the EGS344 line were obtained from the Versailles collection of *Arabidopsis* T-DNA transformants (http://publiclines.versailles.inra.fr/). The meiotic phenotype of all lines was described in [37]. The following mutants were used: *met1-1* [55], *cmt3-11* (SALK_148381, [56]) and *nrpd1a-4* (*pol iv*; SALK_083051, [57]). Plants were grown in a greenhouse (photoperiod 16 h/day and 8 h/night; temperature 20°C; humidity 70%).

### Genetic analyses

The SAIL_904_E06 and the EGS344 lines which are both heterozygous for the *axr1* mutations [37] were crossed to produce F1 hybrid lines. F1s were genotyped to select plants with heteroallelic *axr1*[-/-] mutations or wild-type *AXR1*. Selected F1s were crossed as male to Col-0 to produce BC1 populations. BC1 populations were grown for three weeks and leaf material was collected from rosettes. DNA extraction from 348 plants (BC1 *axr1*[-/-] progeny) and 172 plants (BC1 wild-type progeny) was performed as described in [58]. Genotyping was performed using the KASPAR technology at Plateforme Gentyane, INRA, Clermont-Ferrand, France. The 96 KASPAR markers used in this analysis (S1 Table) are described in [38,59]. They are uniformly distributed on the physical map. Genotyping data were analyzed with Fluigdim software (www.fluigdin.com) with manual corrections. Recombination data were analyzed with MapDisto 1.8.1b [60] with Kosambi parameters. T-tests were carried out to compare wild type and mutant data. Multiple t-test corrections were made using the Benjamini, Kiegel and Yekutieli procedure with a FDR threshold set at 1%. The relative distance from the centromere was calculated as the following: for the North chromosome arm (position of the middle of the interval in Mb)/(length of the chromosome arm in Mb). For the South chromosome arm ((position of the Telomere in Mb)–(position of the middle of the interval in Mb)/(length of the chromosome arm)). r = (number of plants having recombined in the interval)/(total number of plants genotyped for this interval).

### Cytological procedures

Chiasma localization was estimated on metaphase I spread PMC chromosomes counterstained with DAPI based on bivalent configuration as described in [46].

Preparation of intact meiocytes for immunocytology was performed as described in [41]. Antibodies used for immunolocalisation were Guinea-Pig anti-ASY1 (1 in 250 dilution) [41], rabbit anti-DMC1 (1 in 20 dilution) [61], mouse monoclonal anti-H3K9me2 (mAbcam1220), Alexa 568 Goat anti Guinea pig (Life technologie, A11075), Alexa 647 Goat anti mouse (Molecular Probes Invitrogen, A21235) and Alexa Fluor 488 Goat anti rabbit IgG (H+L) Superclonal (Thermoscientifique A27034). Observations were made using a Leica TCS SP8 AOBS confocal imaging. Optical sections (z-step: 0.13μm) were collected using a 100x / 1,4 N. A oil-immersion objective lense. Excitation wavelength for DAPI was 405 nm and emission was collected from 424 to 480 nm. For Alexa 568, the excitation wavelength was 561 nm and emission was collected from 581 to 627 nm. For Alexa 488, the excitation wavelength was 488 nm and emission was collected from 501 to 548 nm. For Alexa 647, the excitation wavelength was 633 nm and emission was collected from 650 to 695 nm. For Alexa 647 and 488 hybrid

detectors were used. Images were processed using the 3D-deconvolution module of Leica LAS X Life Science Microscope Software for Life Science LASX.

FISH on intact meiocytes was performed as described in [41]. The following probes were used: pTAt4 (telomeres) [62], T4P13 and T5J17 (distal regions of chromosomes 3 and 4, respectively, TAIR accessions 3601011 and 3601530, respectively, Fig 1B), K16N2 and F25E4 (proximal regions of chromosomes 3 and 4, respectively, TAIR accessions 1584 and 3601413, respectively, Fig 1B). Telomeres were labelled with digoxigenin and distal and proximal probes were labelled with biotin as described in [63]. The primary antibody mouse anti-DIG (1:100; Roche) and/or Avidin-Texas Red (1:100; Vector Laboratories) were applied as described in [41]. Finally the secondary antibodies, rabbit anti-mouse Alexa-488 (1:100; Molecular Probes) and/or goat anti-avidin–biotin (1:100; Vector Laboratories) were applied. Observations were made using a Zeiss AxioImager 2 microscope (www.zeiss.com). Photographs were taken using a Zeiss AxioCam MR camera driven by Axiovision 4.7 with a 60 9 /1.42 oil objective lens with 1.59 auxiliary magnification at 0.24 µ intervals along the z-axis. All images were further processed with ImageJ Fiji. Selected images were deconvoluted by first generating a theoretical point spread function (PSF) using the 'diffraction PSF 3D' plugin (http://imagej.net/Diffraction_PSF_3D) followed by deconvolution using the 'Iterative Deconvolve 3D' plugin (http://imagej.net/Iterative_Deconvolve_3D).Telomere bouquet volume was measured as described in [41]. Telomere bouquet volume was measured as described in [41].

Preparation of prophase stage spreads for 5-methyl cytosine (5mC) immunocytology was performed according to [64], with the modifications described in [65]. Slides of prophase stage spreads were then dried for 30 minutes at 60˚C. These were then fixed in 1% PFA for 10 minutes and dehydrated through a series of 70%, 90% and 100% EtOH for two minutes in each solution. 50 µl of HB50 were applied onto each slide and were then incubated two minutes at 80˚C for denaturation. Slides were then washed twice in ice-cold 2xSSC (5 minutes each wash), incubated in 100 µl blocking buffer (3% BSA in 1XPBS + 0.1% Tween 20) and washed three times in PBS (5 minutes each wash) and once in TNT for 5 minutes. 50 µl of 5-mC antibody (mouse anti 5-mC 33D3; 1:400; Diagenode) in blocking buffer was pipetted on each slide, parafilm was placed on top and the slides were placed in a moist chamber at 37˚C for 45 minutes. Slides were then washed in TNT three times (5 minutes each wash). 50 µl of secondary antibody rabbit anti-mouse Alexa-488 (see above) was applied, parafilm was placed on top and slides were incubated 45 minutes at 37˚C in a moist chamber. Finally, slides were washed in TNT three times (5 minutes each wash) and mounted in DAPI (2µg/ml in Vectashield). To measure the length of the 5-mC signal on spread chromosomes the freehand line tool in Fiji ImageJ was used to draw along the length of the signal. The "Measure" function in the "Analyze" section allowed us to measure the length of the signal. In the case of the wild type, measurements were performed on an equivalent number of zygotene and pachytene nuclei. In $axr1^{-/-}$, given that there are no pachytene nuclei, measurements were made on zygotene and pachytene-like nuclei.

### DNA methylation analyses

For somatic cells, methylome sequencing, bisulfite treatment, library preparation and whole-genome sequencing were performed by the BGI (China) using HiSeq technology (*Illumina*) producing 150 bp paired-end reads (S2 Table). Reads were trimmed with *Trim_Galore* (*Babraham Bioinformatics*) and aligned to the Col-0 *Arabidopsis thaliana* TAIR10 reference genome with *Bismark* version 0.22.3 (*Babraham Bioinformatics*) and standard default options (*Bowtie2*; 1 mismatch allowed). Identical pairs were collapsed. Only cytosines covered by at least five reads were included to determine the average methylation levels. Differentially Methylated

Regions (DMRs) were identified using the following R packages: *bsseq* version 1.7.7 [66] and *DSS* version 2.11.3 [67] as previously described [68] using standard *DSS* parameters (DMR length > 50 bp, number of Differentially Methylated Loci > 3, more than 50% of C sites with p-value < 0.0001). DMRs closer than 50 bp were merged. To define hypo- or hyper-DMRs, we applied an additional cutoff to keep DMRs with at least a 10% change in methylation ratio for CHHs, 20% for CHGs and 30% for CGs. The BS-seq data [69] used to compare *axr1*<sup>-/-</sup> DMRs with *met1*<sup>-/-</sup>, *cmt3*<sup>-/-</sup>, *kyp*<sup>-/-</sup>, *pol iv*<sup>-/-</sup> or *cmt2*<sup>-/-</sup> DMRs are available from the GEO database under the accession number GSE39901.

Male meiocytes of prophase I were isolated as described [31]. Meiocyte bisulfite sequencing libraries were prepared as described [43]. Reads were processed as for somatic cells except that the—*non_directional* option was used with *bismark*.

## Supporting information

**S1 Fig. Distribution of the number of COs per chromosome.**
(PDF)

**S2 Fig. Pairing behaviour of distal and proximal regions in Col-0 wild type and *axr1*<sup>-/-</sup> during G2 and early prophase.**
(PDF)

**S3 Fig. Co-Immunolocalization of DMC1, ASY1 and H3K9me2 on intact male meiocytes in wild type and *axr1*<sup>-/-</sup> mutant at leptotene.**
(PPTX)

**S4 Fig. Characterization of DAPI dense and NOR regions observed in cytology.**
(PDF)

**S5 Fig. Profiles of methylation in meiocytes.**
(PDF)

**S6 Fig. Profiles of methylation in somatic cells.**
(PDF)

**S7 Fig. Heatmaps visualizing the enrichment of methylation over TEs.**
(PDF)

**S8 Fig. Patterns of methylation in TE families of *axr1*<sup>-/-</sup>.**
(PDF)

**S9 Fig. Profiles of methylation for regions hypermethylated in *axr1*<sup>-/-</sup> mutants.**
(PDF)

**S10 Fig. Localization of hyper Differentially Methylated Regions (DMRs) identified in the *axr1*<sup>-/-</sup> mutant.**
(PDF)

**S11 Fig. Chromosome configuration at Metaphase I of *pol iv*<sup>-/-</sup> and *cmt3*<sup>-/-</sup> meiocytes.**
(TIF)

**S1 Table. Recombination data.**
(XLSX)

**S2 Table. Whole-genome bisulfite sequencing statistics.**
(XLSX)

**S3 Table. HyperDMRs found in *axr1*⁻/⁻ leaves compared to wild type.**
(XLSX)

**S4 Table. HypoDMRs found in *axr1*⁻/⁻ leaves compared to wild type.**
(XLSX)

## Acknowledgments

The authors wish to thank Cécile Raynaud, Eric Jenczewski, Rajeev Kumar, Raphaël Mercier and Jean Molinier for critical reading of the manuscript.

## Author Contributions

**Conceptualization:** Mathilde Grelon, Nicolas Bouché, Christine Mézard.

**Data curation:** Nicolas Christophorou, Jincheng Long, Mathilde Grelon, Xiaoqi Feng, Nicolas Bouché, Christine Mézard.

**Formal analysis:** Nicolas Christophorou, Jincheng Long, Mathilde Grelon, Xiaoqi Feng, Nicolas Bouché, Christine Mézard.

**Funding acquisition:** Xiaoqi Feng, Nicolas Bouché, Christine Mézard.

**Investigation:** Nicolas Christophorou, Wenjing She, Jincheng Long, Aurélie Hurel, Sébastien Beaubiat, Yassir Idir, Marina Tagliaro-Jahns, Aurélie Chambon, Victor Solier, Daniel Vezon, Mathilde Grelon, Xiaoqi Feng, Nicolas Bouché, Christine Mézard.

**Methodology:** Nicolas Christophorou, Xiaoqi Feng, Nicolas Bouché.

**Project administration:** Christine Mézard.

**Resources:** Mathilde Grelon, Xiaoqi Feng, Nicolas Bouché, Christine Mézard.

**Supervision:** Mathilde Grelon, Xiaoqi Feng, Nicolas Bouché, Christine Mézard.

**Validation:** Nicolas Christophorou, Wenjing She, Jincheng Long, Aurélie Hurel, Sébastien Beaubiat, Yassir Idir, Marina Tagliaro-Jahns, Aurélie Chambon, Victor Solier, Daniel Vezon, Mathilde Grelon, Xiaoqi Feng, Nicolas Bouché, Christine Mézard.

**Writing – original draft:** Nicolas Bouché, Christine Mézard.

**Writing – review & editing:** Mathilde Grelon, Xiaoqi Feng, Nicolas Bouché, Christine Mézard.

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
