## [Decision Letter · Decision Letter 0]

23 Aug 2019

Dear Dr Mezard,

Thank you very much for submitting your Research Article entitled 'Meiotic Crossover localisation and DNA methylation are uncoupled In Arabidopsis thaliana when AXR1 is inactivated' to PLOS Genetics. Your manuscript was fully evaluated at the editorial level and by three independent peer reviewers. The reviewers appreciated the attention to an important problem, but raised some substantial concerns about the current manuscript. Based on the reviews, we will not be able to accept this version of the manuscript, but we would be willing to review again a much-revised version. We cannot, of course, promise publication at that time.

If you decide to revise the manuscript for further consideration at PLOS Genetics, please aim to resubmit within the next 60 days, unless it will take extra time to address the concerns of the reviewers, in which case we would appreciate an expected resubmission date by email to plosgenetics@plos.org.

[LINK]

We are sorry that we cannot be more positive about your manuscript at this stage. Please do not hesitate to contact us if you have any concerns or questions.

Yours sincerely,

Ian Henderson

Guest Editor

PLOS Genetics

Gregory P. Copenhaver

Editor-in-Chief

PLOS Genetics

Reviewer's Responses to Questions

**Comments to the Authors:**

Reviewer #1: This manuscript characterizes the distribution of crossovers in an axr1 mutant as well as its methylome and reaches the conclusion they are uncoupled from one another. The crossovers in the mutant become much more distal compared to wild type and pairing is affected, yet telomeric bouquets are unaffected. Using whole genome bisulfite sequencing to explore the DNA methylation landscape in axr1 revealed hypermethylation of transposons especially in the non-CG context.

The data in the axr1 mutant support increased distal crossovers and hypermethylation of transposon sequences, however, given the generality of AXR1 function it seems these are two unrelated studies. The two distinct aspects of the study seemed forced together to make a conclusion that cannot be supported by the data. The results are interesting, but the conclusions should be modified as they as well as the title are not justified by the data. The comments listed below are intended to improve this study with a special emphasis on bisulfite sequencing analysis and interpretation.

Major comments:

1. The presentation of the WGBS data is unconventional and the methods section is lacking detail. How is methylation level determined in table S2 (i.e. C/C+T?)? Same for number of sites (this metric isn’t that useful). How is methylation level in figures determined?

2. In general, there is a lack of context regarding the relationship between DNA methylation and crossovers. It’s my understanding that it’s not DNA methylation per se that affects crossovers but the chromatin it is associated with (i.e. heterochromatin/H3K9me2, etc). As a result the title is inappropriate as there is no evidence presented that the chromatin landscape is altered, as the methylation variation seems to occur at regions that were previously methylated.

3. For all of the metaplots, the authors should include heatmaps, to demonstrate whether there is new methylation occurring at new regions or if it is additional methylation occurring at the same sequences.

4. Although averaging the data between the replicates reveals hypermethylation, especially in the non-CG context, the individual replicates presented alone do not always support this conclusion. Given the authors have replicate data, all replicates should be shown in all figures where appropriate.

5. Figure 3 demonstrates that hypermethylation is occurring in the heterochromatin enriched areas, which are regions where crossovers are suppressed. It shows that there are two unrelated processes ongoing here and makes it hard to understand the study as presented.

6. The cmt3 mutant data shows exactly what is expected, if you lose the CHG methyltransferase you will lose CHG methylation. The nrpd data is odd as it is well known that hypomethylation occurs in this mutant especially in the euchromatic regions, however in the nrpd1 mutant the CHH methylation is higher than many other genotypes including wild type. How is this possible?

7. The last paragraph of the results needs to be rewritten as the comparison to met1 mutants and the mutants in this study are misleading. In met1 mutants, CG methylation is lost, but importantly there is a loss of chromatin structure as well (especially at heterochromatin). This does not happen in the nrpd1 or cmt3 mutants so the comparisons made should be clarified to make this clearer.

Minor comments:

1. The black box in Figure 2A from an image program is odd. Remove it or use a different image.

2. The figure fonts/style etc are different between different figures. Many figures are in low resolution.

3. WGBS conversion rates are not listed in Table S2.

4. The methods need details such that the study can be replicated.

Reviewer #2: In their manuscript entitled “Meiotic Crossover localization and DNA methylation are uncoupled in Arabidopsis thaliana when AXR1 is inactivated” Christophorou et al. extend upon their previous characterization of axr1 mutant phenotypes and investigate the connections between AXR1 and DNA methylation. First, the authors use genetics to combine axr1 mutants from different genetic backgrounds, enabling them to assess the effects of axr1 loss on crossover rates and distribution at a much higher resolution than in their previous work. While these experiments are quite elegant and do indeed provide a more detailed view of the effects of the axr1 mutant, they largely confirm previously reported trends--that in axr1 mutants the number of crossovers remain similar to wild-type but their distribution is altered such that they are more clustered and are relegated to the ends of the chromosomes. As these findings do not provide additional insights into the function of AXR1 in meiosis, they might be more relevant to a highly specialized audience. More interestingly, the authors find that in one axr1 mutant DNA methylation levels are altered. While this of potential interest, especially given previous connections between DNA methylation and meiotic recombination, it remains unclear if these changes in methylation are directly connected to the axr1 dysfunction as only one allele is assessed and no complementation data is provided. Furthermore, this change in methylation doesn’t appear to be connected to the crossover mislocalization effect observed in the axr1 mutants. Thus, to support their claims and elevate the impact of their work, additional experiments (see below) regarding the connections between DNA methylation and AXR1 are required.

Major Comments:

It is not clear what new information is provided in the “DSB initiation occurs normally in axr1-/- meiocytes” section. Perhaps the difference is the inclusion of a marker for H3K9 methylation? If so, this paragraph should be reworded to make this more clear. Alternatively, if no additional insights are provided, this data could be removed.

For the methylation analyses the following additional experiments and analyses should be included to determine what effect, if any axr1 mutants have on DNA methylation.

The bisulfite sequencing should be repeated using another Col allele (for example the one used in Jahns et al 2014) OR bisulfite sequencing using a line complemented with an exogenous AXR1 construct should be included. Without complementation or a second allele, one cannot exclude that the methylation defects are do to a second site mutation in a gene required for DNA methylation.

For the current (and additional) bisulfite sequencing data sets the replicates should be analyzed individually to determine the robustness the hypermethylation phenotypes as this would reveal if all the replicates show hypermethylation at the same sets of loci. The coverage of the current data should be sufficient to allow such overlaps to be conducted.

For Figure S8:

more clones (15-20 per genotype) should be sequence to draw conclusions from this data.

In addition, the single mutant controls should be included in this analysis otherwise its not clear which mutation is influencing the DNA methylation.

Finally, I am not sure what point is being made using this data as it is already well established that met1 mutants affect CG methylation. Please clarify.

Minor comments:

-) Typos in author summary

talso=(also

evnts=events

od=of

-) Figure 1C, the black boxes on the x axis to denote the centromeres are missing

-) Figure 3G legend should refer to the clusters defined in F, not b.

-) Figure S4 the boxed regions showing the DMRs are missing.

-) For Figure S6 what are the scales for the Y axis?

Reviewer #3: The overall premise of the Christophorou et al. manuscript is very attractive as factors controlling the distribution of crossovers (COs) are still poorly understood. The work is a follow-up of a study which found biased distribution of CO events along chromosomes in the axr1 mutant. However, the axr1 phonotype turns out to be quite complex. The mutant exhibits increased DNA methylation genome-wide but this does not appear to be directly associated with the CO distribution patterns. The over conclusion is quite interesting, implying that factors other than chromatin state strongly influence crossover distribution. However, the validity of this tenet may be questioned given how limited information the authors have on the actual DNA methylation patterns during meiosis.

The majority of the information on DNA methylation patterns that Christophorou et al. rely on comes from somatic cells. Taking into account the recent studies on the dynamic nature of DNA methylation during meiosis in Arabidopsis, how indicative are DNA methylation data from leaf tissue for what is happening in meiocytes? For the axr1 single mutant, the authors tried to confirm hypomethylation of meiotic chromosomes using immunolocalization, but this approach is very imprecise. Conducting bisulfate sequencing on meiocytes, or at least meiotic anthers, would clarify this issue.

There is even more uncertainty with regard to the double mutant analyses. Does combining DNA methylation mutations with axr1 indeed bring the DNA methylation patterns back to those of wild-type plants? Christophorou et al. examined DNA methylation by examining it at selected sites in the genome, predominantly TE regions. However, these are not the places where COs are generally formed. To be able to draw meaningful conclusions, the authors should conduct a full-genome analysis.

Finally, the overall conclusion presented in the title that CO localisation and DNA methylation are uncoupled in the axr1 mutant is also uncertain. Have the DNA methylation mutations used in the study completely obliterated DNA methylation in the mutants? DNA sites that are critical for producing specific recombination patterns may still be methylated in the mutant even if there is indeed a global reduction of DNA methylation. A more precise conclusion would be that CO localization in axr1 becomes uncoupled from the effects of the MET1, CMT2, and POLIV genes.

Other major issues:

Page 6: Can the authors use some kind of a statistical measure to assess the degree of clustering? Otherwise, it is a bit difficult to define how strong this phenotype actually is.

Page 8: Is chromosome pairing abolished in the axr1 mutant or just delayed? If the latter, perhaps the defect is in the speed of meiosis progression rather than chromosome interactions.

Page 14/line 5: since the authors only assess distribution of DSBs by a visual assessment of DMC1 foci on immunolocalization images, the statement that the axr1 mutation does not affect DSB formation should be tone-down. Alternatively, can the authors measure spacing of DMC1 foci?

Page 16/line 2: Similarly, I’m not sure that visual assessment of DAPI staining patterns is indeed a good marker of chromosome structure. Can the authors confirm this conclusion by conducting specific measurement of the DAPI-stained regions?

Page 16/line 15: But the study of Underwood et al. seems to contradict this statement…

Page 15/line 21: Is it really true that “DNA methylation down-regulates CO formation…” given that some studies reported decreases of CO numbers in pericentromeric regions of chromosomes in Arabidopsis CG methylation mutants.

Minor issues:

Line 14 of abstract: use “limiting factor” instead of “limitator.”

Lines 10 and 14 of Author Summary: use “localization change” instead of “delocalization.”

Line 12 of Author Summary: use “changes of the pattern” instead of “modification in the pattern.”

Page 1/line 3: use “DNA replication” instead of “replication.”

Page 1/line 25: it would be good to distinguish between species with small genomes, such as Arabidopsis, in which pericentromeric regions are small, and large genome-species, such as barley or wheat, in which the pericentromeric regions are very extensive.

Page 2/line 4: [8] is a wrong citation

Page 2/line 24: “DSBs appear gradually along the genome” – in which species?

Page 3/lines 1-3: “In both human and barley….” But citation [18] only refers to barley.

Page 3/line 15: since the “chromosome mobility machinery” has not been studies in Arabidopsis, this statement may be premature.

Page 4/lines 1 – 2: use “cytosines methylated at non-CG sites” instead of “non-CG methylated cytosines.”

Page 6/line 24: telomere bouquet instead of “bouquet telomeres.”

**Have all data underlying the figures and results presented in the manuscript been provided?**

Reviewer #1: Yes

Reviewer #2: Yes

Reviewer #3: Yes

PLOS authors have the option to publish the peer review history of their article (what does this mean?). If published, this will include your full peer review and any attached files.

Reviewer #1: No

Reviewer #2: No

Reviewer #3: No

---

## [Decision Letter · Decision Letter 1]

20 Jan 2020

Dear Christine,

Thank you very much for submitting your Research Article entitled 'Meiotic crossover localization changes are uncoupled from the effects of MET1, CMT3, or POL IV when AXR1 is inactivated in Arabidopsis thaliana' to PLOS Genetics. 

Your manuscript was fully evaluated at the editorial level and by the 3 original independent peer reviewers. While all three reviewers acknowledge that the revised manuscript is improved, they universally felt that some of their original concerns have not been fully addressed, both in terms of analysis and the use of appropriate controls.

Based on the reviews, we will not be able to accept this version of the manuscript, but we would be willing to review again a revised version that fully addresses these remaining concerns.

If you decide to revise the manuscript for further consideration at PLOS Genetics, please aim to resubmit within the next 60 days, unless it will take extra time to address the concerns of the reviewers, in which case we would appreciate an expected resubmission date by email to plosgenetics@plos.org.

[LINK]

We are sorry that we cannot be more positive about your manuscript at this stage. Please do not hesitate to contact us if you have any concerns or questions.

Yours sincerely,

Ian Henderson

Guest Editor

PLOS Genetics

Gregory P. Copenhaver

Editor-in-Chief

PLOS Genetics

Reviewer's Responses to Questions

**Comments to the Authors:**

Reviewer #1: I appreciate the authors attempt to improve aspects of this manuscript. However, there are still a number of suggestions from my original review that have not been sufficiently addressed.

Table S2 is still quite unconventional to the field although it has improved. Most labs are not interested in then percent of cytosines methylated, as these authors have reported in table S2. Although this can be a useful metric, the authors should include the methylation level for each context in addition to percent of methylated cytosines.

I find the new title odd. In addition to CO localization being uncoupled from MET1, CMT3, etc it's uncoupled from LOTS of genes. As i stated in my original review, this is a classic pleiotropic mutant. The methylation phenotype occurs independent of the cross over phenotype, yet they are forced together in this study. Rublyation/neddylation likely affects a lot of other proteins, which is why these two distinct phenotypes exist. A more accurate title should be included in a new version of this manuscript.

The heatmaps are an integral component of the results. Metaplots will not address if regions that are already methylated become more methylated as it is a single value that collapses all features into a single line. When the line on a metaplot shifts up or down it clearly indicates a change. However, it is unclear if the change is due to new regions becoming methylated/demethylated or existing regions that are methylated gaining/losing methylation or a combination of both. Incorporating heatmaps provides this exact information. The can be included as supplemental data if the authors feel it would interfere with the main figures.

This statement is not useful, "aberrant CO localization pattern correlates with a drastic DNA hypermethylation of TEs in both meiotic and somatic cells, mainly in the non-CG contexts.” DNA hypermethylation occurs genome-wide. As the authors replied in the previous statement, hypermethylation occurs everywhere. If you take their results as presented, that hypermethylated regions are occurring at previously methylated regions, then the correlation between CO localization and non-CG contexts is just a correlation of TE density. The methylation has nothing to do with COs, at least based on the data presented in this study.

I don't doubt the results presented in this study, but there is a need for significant improvement to the clarity of the study More careful interpretation of the results would also benefit the readers and the relevant research fields.

Reviewer #2: In the revised manuscript, Christophorou et al dealt with many of my more minor concerns, but unfortunately did not include any controls (i.e. WT) for their DNA methylation analyses using a second axr1 allele. Since the observed hypermethylation phenotype is quite subtle, and there can be variations in methylation levels between biological replicates, the lack of a WT control makes the new data hard to interpret. Thus, my concerns regarding the connections between AXR1 and DNA methylation remain. Without connections to DNA methylation the remaining data is largely descriptive and confirmatory of previous studies conducted at lower resolution.

Reviewer #3: The revised manuscript is much improved over the original version. However, I find two major issues remaining:

(i) Are DNA methylation patterns in leaves an appropriate proxy for the methylation patterns in reproductive tissues? I think this issue is critical, given that DNA methylation patterns are known to be dynamic during plant development and the fact the that the overall conclusion of study that CO patterns are uncoupled from the methylome patterns defies expectations. I feel the authors should do bisulfate sequencing from anther DNA or at least from flower buds at appropriate stages.

(ii) I still have an issue with the way the authors have compared DNA methylation levels among genotypes. I understand that the axr1 mutants show hypermethylation at many sites across the genome. But are these sites relevant for CO formation? I think the authors should specifically examine the sites identified as harboring COs in wild-type plants.

**Have all data underlying the figures and results presented in the manuscript been provided?**

Reviewer #1: Yes

Reviewer #2: Yes

Reviewer #3: Yes

PLOS authors have the option to publish the peer review history of their article (what does this mean?). If published, this will include your full peer review and any attached files.

Reviewer #1: No

Reviewer #2: No

Reviewer #3: No

---

## [Decision Letter · Decision Letter 2]

15 May 2020

Dear Dr Mezard,

Thank you very much for submitting your Research Article entitled 'Inactivation of AXR1 impairs meiotic crossover localisation and transposon methylation in Arabidopsis thaliana' to PLOS Genetics. Your manuscript was fully evaluated at the editorial level and by independent peer reviewers. The reviewers appreciated the attention to an important topic but identified some aspects of the manuscript that should be improved.

We therefore ask you to modify the manuscript according to the review recommendations before we can consider your manuscript for acceptance. Your revisions should address the specific points made by each reviewer.

[LINK]

Yours sincerely,

Ian Henderson

Guest Editor

PLOS Genetics

Gregory P. Copenhaver

Editor-in-Chief

PLOS Genetics

Dear Christine,

Thank you for your revised manuscript, which was sent to the three original reviewers.

As you can see from their comments you have substantially addressed many of their concerns.

However, reviewers 2 and 3 in particular highlight some additional areas for improvement that need to be dealt with before we can accept your manuscript.

Please attend to these final concerns and submit a revised manuscript.

Best wishes,

Ian

Reviewer's Responses to Questions

**Comments to the Authors:**

Reviewer #1: The title is more representative of the results.

Reviewer #2: see attached

Reviewer #3: I’m really happy that the authors included DNA methylation analysis of axr1 -/- meiocytes in the revised manuscript. These data, however, introduce a major twist in how the link between DNA methylation and CO patterns in axr1 -/-, if any. can be interpreted.

The authors now show that cytosine methylation in the CG and CHG contexts is not altered in axr mutant meiocytes. They found, however, a change in CHH methylation. Although the level of CHH methylation is relatively low, compared to CG and CHG methylation, the relationship between CHH methylation on its own and recombination has not been studied. Hence, it cannot be excluded that the altered CO distribution in axr1 -/- is actually caused by the CHH methylation level change.

Furthermore, since the new data indicate misalignment between meiocyte and leaf DNA methylation, the effect of double mutants between axr1 and met1, cmt3, as well as poliv on DNA methylation patterns in meiosis remains unclear.

Given all this, the conclusions on CO distribution being uncoupled from DNA methylation in axr1 -/- (page 3, page 12, page 13, page 16) are not well supported. Perhaps it would be better just to focus the manuscript on DNA methylation?

Also, Figures S1, S2, and S3 are missing from the revised manuscript.

**Have all data underlying the figures and results presented in the manuscript been provided?**

Reviewer #1: Yes

Reviewer #2: Yes

Reviewer #3: Yes

PLOS authors have the option to publish the peer review history of their article (what does this mean?). If published, this will include your full peer review and any attached files.

Reviewer #1: No

Reviewer #2: No

Reviewer #3: No

---

## [Editor Report · Decision Letter 3]

29 May 2020

Dear Christine,

We are pleased to inform you that your manuscript entitled "AXR1 affects DNA methylation independently of its role in regulating meiotic crossover localization" has been editorially accepted for publication in PLOS Genetics. Congratulations!

Yours sincerely,

Ian Henderson

Guest Editor

PLOS Genetics

Gregory P. Copenhaver

Editor-in-Chief

PLOS Genetics

Comments from the reviewers (if applicable):

**Data Deposition**

http://datadryad.org/submit?journalID=pgenetics&manu=PGENETICS-D-19-01166R3

**Press Queries**

---

## [Editor Report · Acceptance letter]

22 Jun 2020

PGENETICS-D-19-01166R3 

AXR1 affects DNA methylation independently of its role in regulating meiotic crossover localization 

Dear Dr Mezard, 

We are pleased to inform you that your manuscript entitled "AXR1 affects DNA methylation independently of its role in regulating meiotic crossover localization" has been formally accepted for publication in PLOS Genetics! Your manuscript is now with our production department and you will be notified of the publication date in due course.

With kind regards,

Matt Lyles

PLOS Genetics

On behalf of:
